# Playing Non-Professional Football in COVID-19 Time: A Narrative Review of Recommendations, Considerations, and Best Practices

**DOI:** 10.3390/ijerph18020568

**Published:** 2021-01-12

**Authors:** Markel Rico-González, José Pino-Ortega, Luca Paolo Ardigò

**Affiliations:** 1Department of Physical Education and Sport, University of the Basque Country, UPV-EHU, Lasarte 71, 01007 Vitoria-Gasteiz, Spain; markeluniv@gmail.com; 2BIOVETMED & SPORTSCI Research Group, University of Murcia, 30100 Murcia, Spain; 3Department of Physical Activity and Sport Sciences, Faculty of Sports Sciences, International Excellence Campus “Mare Nostrum”, University of Murcia, 30720 San Javier, Spain; 4Department of Neurosciences, Biomedicine and Movement Sciences, School of Exercise and Sport Science, University of Verona, 37131 Verona, Italy

**Keywords:** soccer, SARS-CoV-2, pandemic, season

## Abstract

The coronavirus disease (COVID-19) pandemic in 2020 resulted in widespread interruption of team sports training and competitions. Our aim was to review the recommendations and best practices in return to play in non-professional football after activity lockdown. The authors searched two electronic databases (PubMed, Web of Science) to extract studies published before September 15 2020. Twenty studies explained recommendations, considerations, or best practices in return to play in football, and all of them were clustered into three groups: (1) training load management (*n* = 10), (2) medical recommendations (*n* = 9), and (3) recovery related issues (*n* = 5). The way to establish a progression in training process should be based on training load management and managing the number of stimuli per time. Following the studies, this training process should be divided into three phases: phase 1—physical distancing should be maintained; phases 2 and 3—group training should start. Medical considerations were clustered into different groups: general, pre- and post- training, during training, education, planning to return to competition, and suggestions for post confinement weeks. In particular, social issues, strict hygiene questions, and continuous PCR testing should be considered in return to play over football season. Finally, since a correlation has been found between high-intensive training loads and immunoglobulin A, nutritional and lifestyle recovery strategies should be performed. Moreover, since immunosuppression has been related to congested schedules (<72 h between matches), football federations should avoid this situation.

## 1. Introduction

In late December 2019, a cluster of atypical pneumonia cases in the Chinese Hubei Province of Wuhan resulted in the identification of a novel coronavirus, severe acute respiratory syndrome-coronavirus 2 (SARS-CoV-2), which quickly disseminated to the rest of the world [1,2]. With over 3.3 million confirmed cases worldwide and over 230,000 confirmed deaths, the COVID-19 pandemic is, without a doubt, the single most important event of the 21st century thus far [3]. The coronavirus disease (COVID-19) pandemic has affected all areas of life, and sports is no exception [3].

The COVID-19 pandemic in 2020 has resulted in widespread training and competition interruption of team sports. In return to play, in football, one of the primary concerns has been the maintenance of physical qualities, game specific contact skills, and decision-making ability, attenuating the injury risk on resumption of training and competition [2,4,5,6,7,8]. Hence, sports medicine professionals were called to provide advice on return to training and competitions after the lockdown [1]. Additionally, sports scientists provided training recommendations. The challenge was to achieve an interdisciplinary approach to reduce the impact of aforementioned worldwide pandemic.

Reviews represent a field of literature that includes many studies for caregivers. Such reviews provide recommendations to be considered (and eventually followed) when making decisions [9]. Although sports scientist recommendations should be considered with caution, due to the different contexts related to the different sports settings in different countries [10], a summary of their recommendations may provide reference guidelines for coaches and technical staff in team sports. In brief, and despite the greater number of published reviews about sports in general, and nutrition [11,12], rehabilitation and injury prevention [13], conditioning (such as active lifestyle, resistance training and cardiorespiratory issues [14,15,16]), or general recommendations for return to athletic play during COVID-19 [17], most of them are not methodically performed. Among the reviews, most of them focus on medical issues [16] or healthcare [13], without any specific references to football. However, to the authors’ knowledge, there are not any narrative reviews on recommendations, considerations, and best practices regarding return to play in football.

Among the articles that focused on football, some authors, such as Hamilton et al. [2] and Herrero-González et al. [18] made suggestions about medical issues related to daily practice in football, whereas Bisciotti et al. [1] and Mohr et al. [19] made indications about training load management after activity lockdown. Since there was declared a lack of available scientific evidence and understanding on the novel virus [8], the summarization of these suggestions may be of interest, mainly for non-professional clubs, who, as opposed to professional and semi-professional leagues, have not restarted their training/competition activities as yet. Hence, here, we present recommendations, considerations, and best practices that could be of interest to researchers and other professionals involved in non-professional football. Although some recommendations are based on little data, and given that sharing obtained information could practically contribute toward reducing the gap between theoretical frameworks and daily practices [10], this article could be a reference (at least while waiting for future experimental studies). Therefore, our aim was to review the current recommendations and best practices on return to play in non-professional football after the COVID-19 lockdown.

### Experimental Approach to the Problem

Due to the lockdown affecting the football industry, non-professional players have been constrained to a long period without training stimulus, suffering from a detriment in performance capacities (e.g., tactical, technical, and conditional), increasing the risk of injury incidence. In this sense, coaches have faced exceptional situations, such as reduced time between return to play (football) and competition restart.

In this sense, this article is aimed at establishing a theoretical framework, summarizing a large amount of existing recommendations, considerations, best practices, and training designs, highlighting some issues to be considered by head strength and conditioning coaches, players, and remaining club personnel, implied in daily activity. These issues could be of interest to warrant optimal player performance, reduce injury incidence, and, mainly, ensure football training as a safe environment (i.e., by not contributing to the spread of SARS-COV-2).

## 2. Method

The preferred reporting items for systematic reviews and meta-analyses [20] was followed to perform the search. The procedure carried out for data identification, selection, and extraction is presented in Figure 1. The protocol was not registered prior to the initiation of the project to avoid temporary delays in this critical situation.

### 2.1. Data Sources

The search was performed by two authors to identify articles published before 15 September 2020, in two electronic databases (PubMed and Web of Science) before 11:00 p.m. The authors of this review were not blinded to journal names or manuscript authors. The search was conducted throughout the full text. The search strategy combined terms covering the topics of sport and pandemic-related words: (“team sports” OR basketball OR futsal OR rugby OR soccer OR football OR handball OR hockey) AND (SARS-CoV-2 OR COVID-19). Search strategy focused on any sports, but many of the initially retrieved records referred to football and, therefore, the review finally focused on this sport.

### 2.2. Data Selection

After completion of the search, results were compared between researchers to ensure that the same number of articles was found. Then, one of the authors (M.R.) downloaded the main data from the articles (titles, authors, dates, and databases) to an Excel spreadsheet (Microsoft Excel, Microsoft, Redmond, WA, USA) and removed duplicate records. Subsequently, the same authors screened the remaining records to verify the inclusion/exclusion criteria using a hierarchical approach in two phases. The papers were excluded when they were not original and met the following exclusion criteria: (1) no football or COVID-19 related documents; (2) not about recommendations, considerations or best practices; and (3) not about recommendations or best practices concerning returning to play.

### 2.3. Data Collecting

The considerations, recommendations, or best practices were summarized in Table 1, Table 2 and Table 3 and Figure 2. Specifications were provided when the data of several studies were provided in the discussion and conclusions.

## 3. Results

A total of 73 documents were initially retrieved from the aforementioned databases, of which 17 were duplicated. Thus, a total of 56 articles were downloaded. After screening the titles and abstract against criterion 1 where applicable, and the full text of the remaining papers against criterion 1:14 studies did not meet the inclusion criteria. From the 42 articles, which aim to explain any point of COVID-19 and the team sport, 26 were ruled out because the articles were not about recommendations, considerations, or best practices (criterion 2). Finally, 16 articles were analyzed, and five of them did not fulfil the inclusion criterion 3. Due to the high number of published documents, based on the article per month ratio, documents published after the research date were included (*n* = 1). So, finally, 12 studies were included for the qualitative analysis (Figure 1).

### 3.1. Assessment of Methodological Quality

No quality assessment was done due to the descriptive nature of the studies included. All 13 articles outlined in Table 1, Table 2 and Table 3 and Figure 2 were assessed for suitability and evaluated by authors prior to inclusion. All studies had to meet all items on the criteria list to be included in the analysis.

### 3.2. Study Characteristics

Training recommendations, considerations, and best practices were: training process, medical considerations, and recovery-related recommendations (lifestyle, nutritional, and federation issues).

Recommendations regarding the training process are presented in Table 1 and summarized the Table 2. Table 2 was composed by summarizing the ideas from 10 articles [1,2,8,18,19,21,22,23,24,25].

Medical considerations are presented in Table 1 and summarized in Table 3. It was made from nine articles [1,2,3,8,18,19,21,25,26]. They were divided into general comments, before training recommendations, during training recommendations, after training recommendations, education issues for the players, recommendations to return to official matches, and post confinement (pre-first training) recommendations (see Table 3).

Recovery related recommendations were clustered into lifestyle, nutritional, and some issues related to federation decisions (Table 1 and Figure 2). Lifestyle and nutritional issues were summarized from three articles [2,19,24] and federation issues from another three [18,19,21]

## 4. Discussion

The main aim of this narrative review was to summarize the recommendations and best practices in return to playing football after the COVID-19 lockdown. Although this article may be useful for football in general, since professional and semi-professional leagues have already started, it will be more suitable for young and amateur teams, which continue without activity restarts. Since the COVID-19 pandemic has led players to perform in a pre-season period under an uncommon, congested calendar, a training programming should be done with caution. In this sense, a multidisciplinary approach should be performed, considering the medical recommendations as a main basis. In addition, football federations should help players and clubs avoid congested fixture schedules, which could lead to immunosuppression, and, subsequently, player vulnerabilities to virus contagion. Collected recommendations have been clustered into three groups: (1) training load management, (2) medical recommendations, and (3) recovery considerations, narrowly related with immunosuppression.

### 4.1. Training Load Management

The novel perspectives have led to understand football, in general, and football-specific actions, in particular, as those responses are conditioned by a decision (tactical dimension), a motor skill (technical dimension), which require movement (conditional dimension), and, player psychological states (psychological dimension) [27]. These dimensions may be more accurately extended in different structures identifiable in football performance: bioenergetic, cognitive, coordinative, conditional, creative, socio-affective, emotional-volitional, and mental structures [28]. Since the COVID-19 pandemic has induced competition delays, and pre-season periods are expected to be reduced, the challenge lies in achieving an interdisciplinary approach to reduce the impact of the aforementioned worldwide pandemic [2]. This approach means to develop a training process that adapts to the specific requirements of football, based on specificity, and respecting the different structures that comprise human athletes (being the ones who do sports) and their expressions in motor actions [28]. Based on this approach, the specific competitive environment of football will be facilitated to establish a relationship among the different structures and their organization [28,29].

The first steps are conditioned by the contradictory fact between physical distancing suggestion and football’s social-nature condition. In the first weeks, bioenergetic and conditional dimensions should be separated from the others due to the mandatory situation in which the training task should be decontextualized. Based on different authors, these dimensions should be programmed from greater training volumes to greater intensity training sessions through progressive training processes [1,2,8,21]. Despite the fact that this dynamic is habitually considered in pre-season periods [30], a more congested COVID-19 pre-season could lead the team staff to program an excessive week-to-week training load progression. Following the articles, the weekly training load progressions should not be greater than 10–20%, mainly, to avoid the injury risk [21], which has become a main challenge in the post-COVID-19 season [1,8,18,19,22,23].

So, as a practical viewpoint, the development of aerobic capacity and its subsequent adaptations (i.e., circulatory system, endurance performance, nerve and muscle adaptations) have become the first aim of strength and conditioning coaches. Although, this fact should be performed through decontextualized football-specific actions, due to it belong to the first phase of return to training (see Table 2). In addition to each player’s physical fitness and conditioning assessment, the reference value could be a suitable strategy to program training loads. For example, if professional players run 9–14 km (100–150 m/min) [31] at a higher intensity than 70% of HR_max_ [32,33], running most minutes at 80–90% HR_max_ [32], the first training programs could enroll players in 90-min training sessions with continued training efforts near 70% of HR_max_. Taking into account the progressive dynamic of training loads during this pre-season periods, some protocols, such as the football-specific drill proposed by Drust et al. [34] could be programmed in a second step. Nonetheless, together with bioenergetic and conditional dimensions, individually trained coordinative tasks, such as exercises with balls aimed to improve technical actions, and other tasks aimed to develop coordinative, creative, socio-affective, emotional-volitional, and mental dimensions, should also be considered in additional training drills.

Immediately after confinement, players should receive medical assessments, with body temperature assessment, respiratory/cardiovascular screening, blood analysis, and PCR, which, due to approximately five days of median incubation period for COVID-19 [3,35], the PCR should be repeated some days after. However, independently of the reason of the isolation training period, once the team training is started, players should be continuously enrolled in those training scenarios, demanding decision-making, cognitive, and physical skills (coordinative), and dynamic ever-changing space–time interactions between teammates and opponents in relation to the ball [36]. In this training process, creative, socio-affective, emotional-volitional, and mental structures should be the main basis, under the principle of progression, which, in this case, is related to the idea of the amount and the stimulus per time [27]. In other words, the intensity will be related to the idea of principles and sub-principles of play, and, subsequently, the greater amount of stimulus and the higher intensity demands a certain situation [27]. In this sense, the “different learning” named methodology was proposed based on non-linear pedagogy. Different authors have shown that greater complexity (i.e., more stimuli) and, subsequently, greater levels of concentration, may induce greater improvements in player performances [37,38,39,40,41]. During these tasks, the idea of concentration should always be maximum, while the bioenergetic and conditional dimensions should be progressively programmed. Therefore, in this period, training tasks should achieve through the viewpoint of all professionals involved in a team staff (e.g., head coach, strength and conditioning coach, psychological coach), the management of all dimensions using different constraints: (1) psychologic-related constraints (i.e., principles of complexity or emotional load modifications), (2) technical-tactical constraints (i.e., tactical–technical predominant actions), (3) physical constraints (i.e., preference of muscular concentrations), and (4) task constraints (i.e., number of players, space size, length of time, or level of intermittency) [27].

In summary, the delay of return to play after confinement could aggravate those consequences resulted from other short periods of inactivity, such as summer holidays [5]. On the contrary, what was expected, pre-season periods have been congested, suggesting an excessively rapid training load progression. However, the training process should be related with the idea of progression in both senses: tactical/decision making, using the idea of stimulus, and physical/physiological, based on the idea of progressive training loads. All of these progression perspectives should confluence under a multidisciplinary viewpoint, where the teams have an indissoluble set of structures in which a certain modification in one of them will affect the others. In order to ensure that the physical/physiological objectives programmed to each session are achieved, and considering that tracking technologies are uncommon in non-professional football, real-time feedback monitoring may be an interesting strategy to be considered [23]. In this way, the training session may be stopped before finishing the programmed tasks, or to add some drills to complete each player’s physical and physiological aims.

### 4.2. Medical Considerations

Human-to-human close contact has been highlighted as the main mode of SARS-CoV-2 transmission [25]. In fact, when a player is exposed to another infected player´s droplets or aerosols, there is a risk it will enter the lungs through contact with mucosal membranes (i.e., eyes, mouth, and nose) [25]. So, despite the fact that football is an outdoor team sport played in great spaces by players, the nature of the game makes it so that the contact between teammates and opponents remains unavoidable. Such is so that some authors have assessed the number of infection risk episodes during matches [25,42]. For example, Wong et al. [25] showed that, in a 90-min match, the players were, for a long time (i.e., 19-min) under close contact, maintaining an average of 52 episodes of infectious risky behavior [25]. These findings, together with the fact that the virus can remain on surfaces for days [43], makes it mandatory for all medical suggestions to ensure that only safe individuals should enter into sports facilities. Therefore, Carmody et al. [26] suggested that engaging various departments within the club is essential to coordinate a safe and effective environment for athletes to train and play, together with player education, in regards to SARS-CoV-2 principles (regarding infections, social issues, and principles of transmission) [26].

The compendium of ideas reported in the included articles (Table 2), following different governments and football federations (e.g., Croatian [3], Australian [44], UK [45], Italian [46], or German [47]), reveal that social issues, strict hygiene questions and continuous PCR testing should be the main basis. Therefore, triage questions (e.g., body temperature and Fever, Travel, Occupation, Contact, and Clustering (FTOCC) status) and hygiene issues (e.g., face mask (N95, FFP2, FFP3), cleaning, and disinfection) are mandatory steps before entering into training places. During training sessions, to the interdisciplinary confluence between the head coach and strength and conditioning experts, the work of the remaining workers has become crucial. Hence, team staff members should program training programs, avoiding closed spaces in clubs, and other workers should ensure clean sports equipment.

In summary, based on the wide ways of recommendations, we summarized: (1) the club personnel should warrant that training spaces and club facilities are well prepared before players return to training; (2) players should learn the main issues regarding medical recommendations; (3) coaches, strength and conditioning coaches (designing training tasks), and players (respecting medical recommendations during training) should be connected during training processes to ensure optimal environments.

### 4.3. Recovery Recommendations

Since the essential principle of training under health condition is mandatory in a return to play season, different authors have highlighted different recommendations, mainly focusing on recovery strategies, which could be the main basis to ensure an optimal immune system against upper respiratory tract infections (URTI) from virus or pathogens. To date, the most popular URTI is COVID-19, and immunoglobulin A (IgA) is the main effector against URTI [48]. Such is so that some studies have shown the relationship between IgA and URTI [49,50,51].

In sport settings, different studies have shown the narrow relationship between IgA and intensive efforts [2,19,21,25], especially during congested fixtures [52,53,54]. Based on this scientific evidence, some studies included in this review have warned about this fact [2,21,25]. These suggestions have been clustered into three groups: (1) recovery strategies (nutritional and lifestyle), (2) federation decisions (i.e., schedule-related decisions), and (3) training load management (see Section 3.1). Mainly, different authors highlighted that residual fatigue should be between 24–48 h, suggesting up to 72 h of rest time between intensive and prolonged physical exertions [18,25]. This idea is consistent with different studies that have shown decrements in IgA levels until 72 h after intense efforts [48,50,55,56,57]. Moreover, since Putlur et al. [58] found that 82% of illnesses could be explained by preceding decrease in salivary immunoglobulin A, and following Nakamura et al. [59], who found a decrement in IgA three days before URTI, caution is needed in the distribution of high intensity sessions.

Finally, since the stress resulting from poor sleep quality, low mood, and ineffective recovery strategies may all negatively impact an individual’s immune function [2,60], the administration of appropriate interventions based on objective evidence is a mandatory strategy [53]. In this sense, the idea of recovery strategies between matches is highly suitable, even though they are separated by more than 72 h. Specifically, since different authors have related nutritional and lifestyle recovery strategies with immunosuppression [2,11,19], carbohydrate/protein enrichment diets—this, together with compression garments, soft tissue therapies, and sleep hygiene strategies, seem to be suitable strategies to ensure match-to-match and training-to-training efforts.

In summary, although these suggestions are made with caution, football federations should avoid competition programs where matches are separated for less than 72 h. Nonetheless, despite sufficient recovery time between intensity efforts, lifestyle and nutritional strategies should be followed.

## 5. Certainties, Speculations, and Future Lines of Investigation

### 5.1. Training Load Management

It is a certainty that objective data (i.e., quantification of training load) may be helpful to conduct a cautious, pre-season after activity lockdown. However, unlike other activity lockdown situations, such as summer periods, it is speculated that several negative effects caused on athletes may suppose a novel training load management period, in which the number of injuries increase. Hence, further studies should assess training load periodization and share information to the remaining clubs. Beyond each non-professional football club’s economic situation, the rating of perceived exertion (RPE) may be a suitable way to collect and share information.

### 5.2. Medical Issues

Since infected human droplets or aerosols are ways to virus transmission [25], and avoiding physical distancing seems to be the best practice to avoid it, collision sports have been empirically assessed and considered as risk environments [25]. Hence, at initial stages, training should be further to football-specific social structures, which influence collective performance. However, no study has assessed the suitability of tasks with space restrictions, where the pitch is divided into different areas and each player is assigned to one of them, according to specific playing positions, not allowing players to move out during training game [61]. These training tasks could limit social contact, while the players improve their bioenergetic, cognitive, coordinative, conditional, creative, socio-affective, emotional-volitional, and mental structures, even during the first steps of the pre-season.

### 5.3. Recovery Related Issues

It is a certainty that training load management and enough recovery periods are related with the immune system, especially, with IgA. In fact, different studies have shown the effects of very high training loads with IgA [49,53,62,63], and lower levels of IgA with URTI [49,50,51,59]. However a relationship between IgA levels and SARS-CoV-19 is not clear. Further studies should assess if salivary samples may be enough to detect an infected athlete before symptoms appear. However, saliva analysis in non-professional football is not common. For this reason, further studies should corroborate the idea of relationships between IgA and other assessment methods, such as global positioning systems [52,63], or even with RPE [49,62,63], and wellness questionnaires [54].

Finally, it is a certainty that different game durations lead to different player performances [64]. Since some studies speculate about the modification on game duration, and others on the number of substitutions after an activity lockdown season, further studies should assess the impact of these modifications in competition demands.

## 6. Study Limits

Due to the immediate necessity to publish articles that guide sports practices in this critical situation, the number of articles published in a short period of time is very large. Hence, there is the possibility that publications after the search date exist that have not been included in the analysis. Furthermore, literature on different aspects of COVID-19 is accumulating fast and its quality is sometimes dubious. Therefore, this leaves some uncertainty concerning the assumptions made on this matter.

Due to the fact that, at least sometimes recommendation-based studies lack empirical data, supporting current considerations and best practices with further research studies could reduce, at least practically, the gap between the theoretical framework and daily practice.

## 7. Practical Applications

Training sessions should start with warm-up time, when players perform ball drills, and other games improving socio-affective and emotional-volitional, even during first steps, when physical distancing should be maintained. In the main part of the session, coaches should be aware that any actions are unconditionally influenced by a context and, subsequently, the tactical dimension should be mainly considered. In this regard, training task within this part of the training should maintain the highest level of stimulus in order to warrant player concentration. However, tactical dimensions do not exist by themselves; therefore, it makes sense to take them into account only when they occur in interaction with other dimensions. Practically, coaches could consider training tasks within changing environments. Therefore, the players would train, self-adapting to these new circumstances and, simultaneously, face different game constraints (space, time, and number of players) to establish a weekly training load progression. For example, players could perform training tasks, in which boundaries are continuously changing (demand greater concentration) within greater spaces, per player, during the first steps of returning to play football, in order to perform more aerobic efforts. Moreover, players could play within smaller spaces and with greater intensity, demanding tasks as the competition period approaches.

Simultaneously during COVID-19, some strategies should be planned to warrant player predisposition for intensive training sessions or matches, over return to play during football season. In this sense, team staff members should organize theoretical sessions about nutritional and sleep hygiene strategies. On the one hand, coaches should teach players that they should follow a carbohydrate-rich diet with high protein ingestion (0.4 g kg^-1^ per meal), and additional probiotics, viz., vitamin C and D3 supplementation. On the other hand, players should learn the main sleep hygiene strategies: (1) post-training shower, (2) quiet, cool, and dark bed room environment, (3) avoid bedroom clocks, (4) no technology devices 1 h before sleep, (5) avoid caffeine, alcohol, and other stimulants prior to sleep, (6) high-electrolyte fluid after exercise and prior to sleep, and, (7) light, high glycemic index meals with little protein [65,66]). These steps should be followed prior to sleep, especially after late night sessions [67]. In addition, these educational sessions should highlight maintaining physical distancing and strict-hygiene topics, which players should become aware of since they start training and playing for their clubs.

## 8. Conclusions

Non-common, congested pre-season periods have given coaches and technical staff members a supreme challenge, to program training sessions and behavioral prescription, where the main aim is to achieve optimal performance for competition periods, considering the high risk of players’ health. In this sense, a multidisciplinary training approach seems to be the main basis to perform it, where medical recommendations should be the key point, and federations should conduct the best schedule-related decisions to help clubs in this critical situation.

## Figures and Tables

**Figure 1 ijerph-18-00568-f001:**
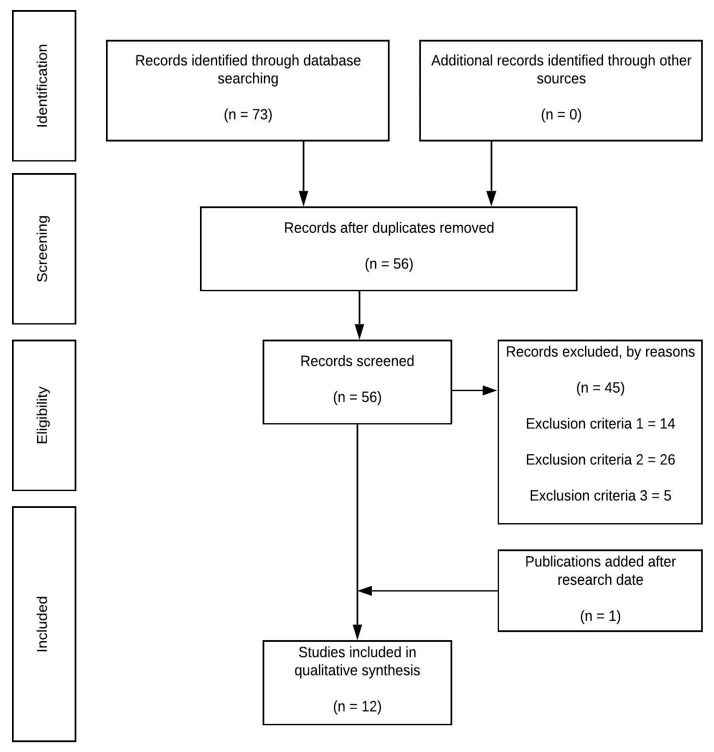
Flow diagram of the study.

**Figure 2 ijerph-18-00568-f002:**
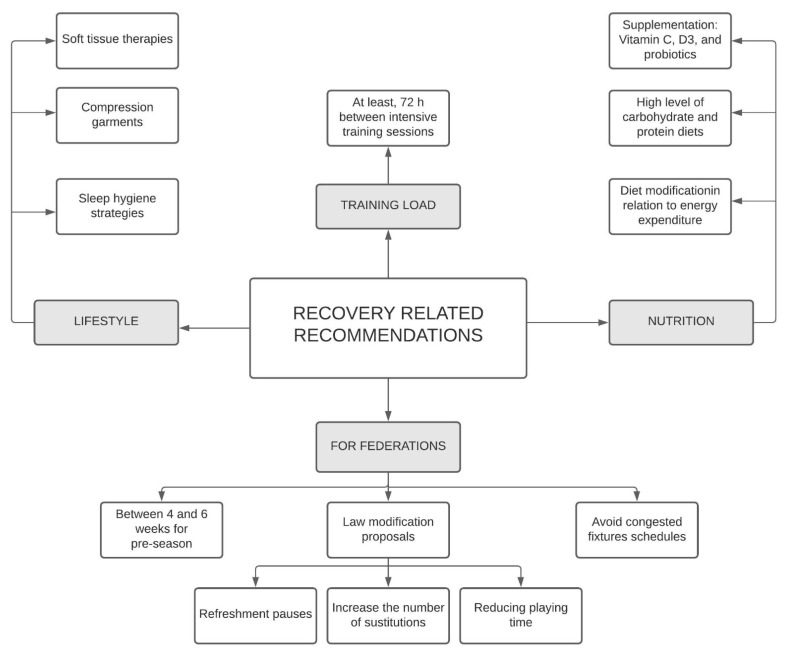
Summary of recovery related recommendations.

**Table 1 ijerph-18-00568-t001:** Articles included and main findings.

Reference	Recommendations about?	Training Sessions (See Table 2)					Medical Issues (See Table 3)			Recovery Strategies (See Figure 2)			
		Aerobic Training	Anaerobic Training	Speed/Strength/Neuromuscular Training	Ball Drills	Injury Prevention Programs/Issues	Social Issues	Hygiene Issues	Test	Nutritional	Wellness	Training Load	Federation Issues
Bisciotti et al. [1]	TL management Injury prevention	Intensity progression (from aerobic to anaerobic training)		Three phases: power production; plyometric; repeated sprints		For joints: multicomponent prevention programs. For neuromuscular control: Single leg exercises			Pre competition medical assessment				
		10 sessions At 70–75% Slow rhythmic	Stimulation of fast twitch fibers. At 50–85% High rhythmic										
Mohr et al. [19]	TL management	Progressiveness of loading From moderate to high intensity, combined with basic power training			Warm up with ball circuit Specificity (SSG)	3–5 weeks pre competition period	2 m of social distance at initial stage			Carbohydrate enrichment diet	Proper sleep Compression garments	Avoid exhaustive TL	Reduce playing time Increase substitutions
Castagna et al. [21]	TL management Guide safe	Maximal and submaximal test Consider the ideal worst match scenario to prepare match psychological and physiological demands Weekly TL progression < 10–20%					Social distancing	Sanitized VAR rooms	Repeated test PCMA			24–48 h recovery time post session	Reduce match duration (30-40 min/half)
Herrero-González et al. [18]	TL management Safety	Phase 1: individualized test (power, endurance, joint mobility and body composition). Phase 2: football specific games including power, endurance, high-intensity intermittent exercise capacity and speed			Football specific training phase				Pre training Temperature Respiratory and cardiovascular screening			72 h between matches	4-6 weeks before competition Refreshment pauses Increase the number of substitutions
Sarto et al. [22]	Awareness in sport programming	Improve physiological and mental functions Involve all stakeholders in the decisions Ensure recovery, specially, during congested fixtures											Allow prolonged pre-season
Nassis et al. [23]	TL management	The use of technology for real time feedback Positional and individual variability in fatigue and recovery patterns should be established											
		High-intensity bouts with emphasis not only on the number of bouts but also on the density too (i.e., number of high-intensity efforts within 1–2 min)											
Hamilton et al. [2]	TL management Safety	Multi-disciplinary planning for reintroduction of training volume and intensity, taking into consideration both individual and squad-based factors			Consider squad-based factors		Open areas Face mask	Alcohol based cleaning products	Triage Temperature NSAID when managing URTI	Adapt diet to a new energy expenditure Consume protein regularly	Sleep quality low mood Soft tissue recovery strategy	Well organized TL	
Stokes et al. [8]	TL management Safety	Six-week training blocks. Assess with RPE and other indicators of reduced cardiopulmonary function. Progression through small increments in intensity (e.g., speed, distance kicked/passed, player numbers) and volume (e.g., nº repetitions, duration, nº of player). Introduction of randomization, reaction time, fatigue and decision making							Symptom free for 7 days and return no sooner than day 10				
					From SGG with small areas to greater								
Huyghe et al. [24]	TL management and recovery	Consider a preliminary baseline with reduced loads upon return to competition											
		2/4 weeks of unstructured, low impact, and low volume activities. 50/30/20/10 rule		Near competition HIIT (87–97% peak HR)						Nutrition education. Carbohydrate ingestion. Quercetin-rich foods			
Wong et al. [25]	TL recovery and safety						Physical distancing	Clean equipment, surfaces. Avoid contact	Temperature Declare FTOCC status			72 h between matches high intensity sessions	
Carmody et al. [26]	Safety						Read COVID-19 guidelines	Educate players. Installation disinfection. No sharing of thingsFace mask.	Any person with symptoms should be at home for, at least, 14 days				
Primorac et al. [3]	Training phases Safety	Return to physical activity in 4 phases: (1) training in small groups (< 5 players); (2) training of the entire team; (3) national league competitions; (4) international competitions					5 m physical distance. Training outdoors.Go home in the same transport	No sharing	Negative consecutive two PCR pharyngeal swabs over a 5-day interval				

COVID-19: coronavirus disease; DLCO: diffusing capacity of the lung for carbon monoxide; FeNO: fractional exhaled nitric oxide; FTOCC: Fever, Travel, Occupation, Contact and Clustering status; HIIT: high intensity interval training; HR: heart rate; NSAIDs: Non-steroidal anti-inflammatory drugs; PCMA: pre-competition medical assessment; RPE: rating of perceived exertion; TL training load; URTI: upper respiratory tract infection/symptoms; VAR: video assistant referee.

**Table 2 ijerph-18-00568-t002:** Summary of recommendations about training process after confinement.

Phase 1		Phase 2		Phase 3	
Sub-Phase 1 (1st Week)		Sub-Phase 1 (3rd Week)		Sub-Phase 1 (5th Week)	
Conditional Dimension	Technic/Tactic Dimension	Conditional Dimension	Technic/Tactic Dimension	Conditional Dimension	Technic/Tactic Dimension
1. Aerobic adaptation (low intensity). 2. Maintain speed and endurance.	Individual ball drills	1. Aerobic-anaerobic training (moderate). 1.1. Intermittent exercise 1. 2. Speed and endurance production. 3. Selective stimulation of the fast twitch fibers. 4. Plyometric training 1.	High-intensity ball drills 1	1. Aerobic-anaerobic training (high-intensity). 1.1. Repeated sprint ability 2. 2. Speed training.	High intensity ball drills 2
1. Aerobic to anaerobic training (from low to moderate). 2. Maintenance and production of speed and endurance.		1. Aerobic-anaerobic training (from moderate to high). 1.1. Intermittent exercise 2. 1.2. Repeated sprint ability 1. 2. From speed and endurance production to speed training. 3. Selective stimulation of the fast twitch fibers. 4. Plyometric training 2. 5. Repeated sprint ability.		RETURN TO PLAYING FOOTBALL	
INDIVIDUAL TRAINING PHYSICAL DISTANCING		GROUP TRAINING FOOTBALL-SPECIFIC TRAINING DRILLS			

**Table 3 ijerph-18-00568-t003:** Summary of medical considerations.

General Recommendations		
- Follow national and international (when travelling) guidelines. - Follow restrictions. - Engage department within a club to coordinate a safe and effective environment. - Seamless communication with colleagues in other sports or clubs with similar risk profiles. - Medications may influence the efficiency of immune function. - A player with a confirmed or suspected COVID-19 infection should be symptom free for 7 days and RTP no sooner than day 10 of the infection.		
**Pre-Training**	**During Training**	**After Training**
- Players and coaches must go alone to training. - Triage questions of all players from a car park (single point of entry). - Measure body temperature and declare FTOCC status. - Cleaning and disinfection. - Face mask (N95, FFP2, FFP3).	- Four phases: (1) training in small groups (respecting 2 m of social distance); (2) training of the entire team; (3) national competitions; (4) international competitions. - Sporting equipment should be cleaned. - No sharing bottles, food, communal indoor facilities. - Training must be performed outdoors (no closed spaces into clubs). - Sprinting and running in different lanes, or 40 m distance between each other (only for initial sessions). - Assess and monitor physiological markers including resting, exercising and recovery.	- Sporting equipment should be cleaned as frequently as possible. A dilution of 1:50 of standard 99 bleach for large settings and 70% ethanol is recommended for smaller surfaces. - Face mask (N95, FFP2, FFP3).
**Education**	**Planning to Return to Match Officials**	**After Confinament**
Educate player in: Principles of infection. Principles of social issues Principles to report symptoms (e.g., consultation using telehealth or one by one).	- Match officials should be preferentially trained in specifically sanitized venues for the entire duration of the remaining part of the season, and repeated COVID-19 tests across the final stages of the competitive season should be considered to warrant personal and other people’s (family members, colleagues, players, and citizens) health. - VAR rooms must be sanitized, and operators enabled to officiate while social distancing, and with face masks and plastic gloves.	Re-training could start after COVID-19 pre-competitive medical assessment: Body temperature. Blood analysis. PCR. Respiratory and cardiovascular screening.

FTOCC: Fever, Travel, Occupation, Contact, and Clustering status; RTP: return to play.

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
