# Peer review of "Playing Non-Professional Football in COVID-19 Time: A Narrative Review of Recommendations, Considerations, and Best Practices"

_ijerph, 2021, doi:10.3390/ijerph18020568_

Round 1

Reviewer 1 Report

Since I'm more familiar with other aspects of sport and exercise and having no expertise in football, I must say that I liked reading the paper. It is easy to follow, understandable and at the same time - sufficiently scientifically rigorous. The approach (PRISM mainly) is clearly presented, and results are easy to follow. I have some minor comments I hope will improve the readability of the paper and applicability of the results.

Abstract

Shorten the study bacground to one sentence, but pay more attention on reporting the results - present more details about how many studies suggested what? Also, I prefer to see some kind of short overview of informations which are neccesarry but not provided so far

Introduction

I suggest authors to shortly overview (eventual) reviews from other fields of sport science which covered similar topics - one paragraph will be sufficient

Discussion

Line 146 - please provide more details on "interdisciplinary approach"

Line 158 onward. Sentence is too long

Line 178-181; OK I may agree with the statement, but it would be good if some emirpircal evidences will be provided. Maybe some previous empirical study which confirmed suggested effects in pre-pandemic time?

Line 206 onward. Sentence needs reference

Author Response

Since I'm more familiar with other aspects of sport and exercise and having no expertise in football, I must say that I liked reading the paper. It is easy to follow, understandable and at the same time - sufficiently scientifically rigorous. The approach (PRISMA mainly) is clearly presented, and results are easy to follow. I have some minor comments I hope will improve the readability of the paper and applicability of the results.

Response: thank you very much for your acknowledgements, and for supporting the work.

Abstract

Shorten the study background to one sentence, but pay more attention on reporting the results - present more details about how many studies suggested what? Also, I prefer to see some kind of short overview of information which are necessary but not provided so far.

Response: Thank you very much for this issue. Your suggestion has been followed.

Introduction

I suggest authors to shortly overview (eventual) reviews from other fields of sport science, which covered similar topics - one paragraph will be sufficient.

Response: your suggestion has been followed.

Discussion

Line 146 - please provide more details on "interdisciplinary approach".

Response: we agree with you. Likely, the paragraph needed a wider explanation. It has been added.

Line 158 onward. Sentence is too long.

Response: the sentence was divided.

Line 178-181; OK, I may agree with the statement, but it would be good if some empirical evidences will be provided. Maybe some previous empirical study which confirmed suggested effects in pre-pandemic time?

Response: Thank you very much for your suggestion. It was very helpful to improve the idea. The concept of differential learning was introduced:

In this sense, the “different learning” named methodology has been proposed based on non-linear pedagogy. Different authors have shown that greater complexity (i.e. more number of stimuli) and, subsequently, greater levels of concentration may induce greater improvements in players performance.”

Line 206 onward. Sentence needs reference.

Response: your suggestion was followed.

Reviewer 2 Report

Dear Authors,

it is very hard to write a review and even harder to extract the sound and valuable recommendations from the data whose quality has not been tested. 

Author Response

Dear Authors,

it is very hard to write a review and even harder to extract the sound and valuable recommendations from the data whose quality has not been tested.

Response: we agree with you. However, a PRISMA checklist is not a quality assessment instrument intended to gauge the worth of a systematic review [1]. In the articles, an additional assessment checklist is presented. Due to the qualitative and quantitative studies that can be used, the quality assessment is not mandatory for all articles. Specifically, studies based on a recognized classification method as the nature of the search are descriptive [2–4]. In this regard, Colomer et al. [2] suggested a table for suitability and evaluation by a panel of authors before inclusion. All studies had to meet every item on the criteria list to be included in the analysis [2].

For this reason, we explain: “No quality assessment has been done due to the descriptive nature of the studies included. All 13 articles outlined in tables 1, 2 and 3 and figure 2 were assessed for suitability and evaluated by authors prior to inclusion. All studies had to meet all items on the criteria list to be included in analysis.”

  1. Moher, D.; Liberati, A.; Tetzlaff, J.; Altman, D.G. Preferred reporting items for systematic reviews and meta-analyses: The PRISMA statement. Int. J. Surg. 2010, 8, 336–341, doi:10.1016/j.ijsu.2010.02.007.
  2. Colomer, C.M.E.; Pyne, D.B.; Mooney, M.; McKune, A.; Serpell, B.G. Performance Analysis in Rugby Union: a Critical Systematic Review. Sports Med. - Open 2020, 6, 4, doi:10.1186/s40798-019-0232-x.
  3. Rico-González, M.; Los Arcos, A.; Nakamura, F.Y.; Moura, F.A.; Pino-Ortega, J. The use of technology and sampling frequency to measure variables of tactical positioning in team sports: a systematic review. Res. Sports Med. 2020, 28, 279–292, doi:10.1080/15438627.2019.1660879.
  4. Rico-González, M.; Pino-Ortega, J.; Nakamura, F.Y.; Moura, F.A.; Arcos, A.L. Identification, Computational Examination, Critical Assessment and Future Considerations of Distance Variables to Assess Collective Tactical Behaviour in Team Invasion Sports by Positional Data: A Systematic Review. Int J Env. Res Public Health 2020, 14.

Reviewer 3 Report

Line 2-4. The title must have the beginning of each word in capital letters.

Abstract

Line 16. Although COVID -19 is a familiar term, scientific writing requires that the first time an acronym appears it be explained. This is the case for COVID 19: 'CO' stands for corona, 'VI' for virus, and 'D' for disease.

Introduction

Line 36. Apply the same as line 16 for SARS (Severe Acute Respiratory Syndrome-related virus)

The introduction does not make it clear whether the search was made on male or female competitions, or both, or from, or until, which age of participants or which categories are referred to as "non-professional football".

Up to line 56 it makes a good general theoretical approximation but without introducing us to the contributions regarding this pandemic and football training which later indicate that they have reviewed. This should be explained in this section of the article in a way that allows the future reader to know what are the different findings produced by the scientific research

Method

Line 70. The two electronic databases used are PubMed and Web of Science and are not an example among others. Please correct it.

Line 74. "SARS-CoV-2" does not need to be in double quotes as it is a single term.

Results

The results, which are excellent and helpful to the coaches, should indicate in the tables what recommendations each of the 13 papers that have been finally selected makes. As tables 1 and 2 are written, the contributions of each of the papers are unknown. This should be clearly identified in each recommendation.

Discussion

The Discussion is undoubtedly the best section of the article and the recommendations grouped in its three sections: training load management, medical recommendations, and recovery considerations, narrowly related with immunosuppression, are very helpful.

However, as it was not done in the introduction, the concept and search for "non-professional football" is still unknown and this ambiguity should be clarified from the beginning of the article as I have asked for in the first corrections.

Finally, the sections on Practical Applications and Conclusions are well explained and are of potential value both to practitioners and to their coaches and assistants, which will surely make this article a source of potential references and quotations in other articles.

Author Response

Line 2-4. The title must have the beginning of each word in capital letters.

Response: your suggestion was followed:

Playing Non-Professional Football in COVID-19 Time: A Narrative Review of Recommendations, Considerations, and Best Practices.

Abstract

Line 16. Although COVID -19 is a familiar term, scientific writing requires that the first time an acronym appears it be explained. This is the case for COVID 19: 'CO' stands for corona, 'VI' for virus, and 'D' for disease.

Response: your suggestion was followed in both abstract and introduction.

Introduction

Line 36. Apply the same as line 16 for SARS (Severe Acute Respiratory Syndrome-related virus).

Response: your suggestion was followed.

The introduction does not make it clear whether the search was made on male or female competitions, or both, or from, or until, which age of participants or which categories are referred to as "non-professional football".

Response: Thank you very much for your issue. It was not detailed, because this recommendation, considerations and best practices could be used overall. In fact, the articles included in the qualitative synthesis are not detailed.

Up to line 56 it makes a good general theoretical approximation but without introducing us to the contributions regarding this pandemic and football training which later indicate that they have reviewed. This should be explained in this section of the article in a way that allows the future reader to know what are the different findings produced by the scientific research.

Response: thank you very much. Your suggestion was followed.

Method

Line 70. The two electronic databases used are PubMed and Web of Science and are not an example among others. Please correct it.

Response: thank you very much. “(i.e. PubMed, and Web of Science)” has been replaced by “(PubMed and Web of Science)”.

Line 74. "SARS-CoV-2" does not need to be in double quotes as it is a single term.

Response: we agree with you. Your suggestion was followed.

Results

The results, which are excellent and helpful to the coaches, should indicate in the tables what recommendations each of the 13 papers that have been finally selected makes. As tables 1 and 2 are written, the contributions of each of the papers are unknown. This should be clearly identified in each recommendation.

Response: Thank you very much for your suggestion. We added the references in the tables 2 and 3. However, following your suggestion, we added a new table 1 instead of adding information in the tables 2 and 3. If the reviewer further think that it is necessary to add references in each recommendations of tables 2 and 3, we will do it.

Discussion

The Discussion is undoubtedly the best section of the article and the recommendations grouped in its three sections: training load management, medical recommendations, and recovery considerations, narrowly related with immunosuppression, are very helpful.

Response: thank you very much for your acknowledgement.

However, as it was not done in the introduction, the concept and search for "non-professional football" is still unknown and this ambiguity should be clarified from the beginning of the article as I have asked for in the first corrections.

Response: although the article could be useful for football in general, the fact that professional and semi-professional football players already restarted the competition could make the article more helpful for those categories that did not started, yet. As you suggest, this fact was specified in the discussion.

Finally, the sections on Practical Applications and Conclusions are well explained and are of potential value both to practitioners and to their coaches and assistants, which will surely make this article a source of potential references and quotations in other articles.

Response: thank you very much for your acknowledgement.

Round 2

Reviewer 2 Report

Dear Authors,

you first decided to do the literature search on the effects of lockdown on professional team sport and tried to summarize the potential guidelines on how to aleviate the reintoducement into trainings and matches. After the key word search you decided upon analyzing data on football. The data on different aspects of COVID-19, as you are aware of, is accumulating enormously fast and its quailty is sometimes doubious. It is therefore difficult to make assumptions and give recommendations for yet another completely different sport such as amateur football. I think there are too many limitations of the work for it to be suitable for publication.

Author Response

You first decided to do the literature search on the effects of lockdown on professional team sport and tried to summarize the potential guidelines on how to alleviate the reintroducement into trainings and matches. After the key word search you decided upon analyzing data on football. The data on different aspects of COVID-19, as you are aware of, is accumulating enormously fast and its quality is sometimes dubious. It is therefore difficult to make assumptions and give recommendations for yet another completely different sport such as amateur football. I think there are too many limitations of the work for it to be suitable for publication.

Response: thank you very much for this viewpoint. These issues were explained and added within previous manuscript revision in introduction, method and study limitation sections. However, following sentences were added to one bullet of 5.    Study limits:

“Furthermore, literature on different aspects of COVID-19 is accumulating hugely fast and its quality is sometimes dubious. Therefore, this leaves some uncertainty space on any assumptions made on this matter.”